# Acute strength exercise training impacts differently the HERV-W expression and inflammatory biomarkers in resistance exercise training individuals

Lucas Vinicius Morais[1], Samuel Nascimento dos Santos[1], Tabatah Hellen Gomes[1], Camila Malta Romano[2], Patricia Colombo-Souza[1], Jonatas Bussador Amaral[3], Marina Tiemi Shio[1], Lucas Melo Neves[1,4], André Luis Lacerda Bachi[1], Carolina Nunes França[1], Luiz Henrique da Silva Nali [1]*

1 Post-Graduation Program in Health Sciences, Santo Amaro University, São Paulo, Brazil, 2 Hospital das Clínicas da Faculdade de Medicina da Universidade de São Paulo (HCFMUSP) LIM-52, São Paulo, Brazil, 3 ENT Research Lab, Department of Otorhinolaryngology-Head and Neck Surgery, Federal University of Sao Paulo, Sao Paulo, Brazil, 4 Bipolar Disorder Program (PROMAN), Department of Psychiatry, Medical School, University of São Paulo, São Paulo, Brazil

* luiznali@gmail.com, lnali@prof.unisa.br

**Data Availability Statement:** All relevant data are within the manuscript and its Supporting Information files.

## Abstract

### Background

Human Endogenous Retroviruses (HERVs) are fossil viruses that composes 8% of the human genome and plays several important roles in human physiology, including muscle repair/myogenesis. It is believed that inflammation may also regulate HERV expression, and therefore may contribute in the muscle repair, especially after training exercise. Hence, this study aimed to assess the level of HERVs expression and inflammation profile in practitioners' resistance exercises after an acute strength training session.

### Methods

Healthy volunteers were separated in regular practitioners of resistance exercise training group (REG, n = 27) and non-trained individuals (Control Group, n = 20). All individuals performed a strength exercise section. Blood samples were collected before the exercise (T0) and 45 minutes after the training session (T1). HERV-K (HML1-10) and W were relatively quantified, cytokine concentration and circulating microparticles were assessed.

### Results

REG presented higher level of HERV-W expression (~2.5 fold change) than CG at T1 (p<0.01). No difference was observed in the levels of HERV-K expression between the groups as well as the time points. Higher serum TNF-α and IL-10 levels were verified post-training session in REG and CG (p<0.01), and in REG was found a positive correlation between the levels of TNF-α at T1 and IL-10 at T0 (p = 0.01). Finally, a lower endothelial microparticle percentage was observed in REG at T1 than in T0 (p = 0.04).

**Funding:** Fundação de Amparo à Pesquisa do Estado de São Paulo (FAPESP), grants # 2023/08773-0 and 2020/11619-5

**Competing interests:** The authors have declared that no competing interests exist.

## Conclusion

REG individuals exhibited a significant upregulation of HERV-W and modulation of inflammatory markers when compared to CG. This combined effect could potentially support the process of skeletal muscle repair in the exercised individuals.

## Introduction

Inflammation is a vital and ubiquitous conserved biological process that involves the activation of immune and non-immune cells with the goal that not only guaranteeing host defense against pathogenic agents and toxins but also restoring tissue homeostasis in response to cell/tissue injury. In particular, acute inflammation is characterized by events and releasing molecules well-coordinated and self-controlled that is maintained as long as the threat/injury has not been totally eliminated/solved, disappearing after these events have passed [1, 2]. Interestingly, it is recognized that an exercise training session can trigger an acute inflammatory response in which its extent depends on both intensity and duration [3]. In this sense, the magnitude of the inflammation acutely elicited by exercise training can be assessed through the levels and expression of different inflammatory biomarkers [4].

Among some biomarkers, IL-6 is the most studied molecule released from contracting skeletal muscle during physical exercise performance and presents a wide range of actions, which includes its ability to: (1) promote lipolysis in adiposity tissue and glycolysis in the liver; (2) improve glucose uptake by muscle during the exercise session; (3) stimulate the multiplication of satellite cells and driving hypertrophy of the muscle; (4) induce the increase the systemic levels of the anti-inflammatory cytokine IL-10, leading to a control of inflammation generated by the physical exercise performance, among other actions [3].

Another cytokine that rises promptly in the bloodstream in response to muscle damage related to exercise training is the Tumor Necrosis Factor-alpha (TNF-α). This factor can be released by different cell types, especially circulating monocytes, being particularly noteworthy contributors [5]. Concerning the literature, this cytokine presents several regulatory actions in inflammation and tissue injury, and, although it can be involved in the induction of both necrosis and apoptosis of myocytes. And, particularly in chronic inflammatory situations [6] the TNF-α is able to regulate the myogenesis and muscle regeneration [7, 8].

Beyond these aspects, it is postulated that the inflammation condition seems to impact the transcription of Human Endogenous Retrovirus (HERVs). They have fixed within the genome of their germline cells, and during the process of endogenization HERVs were passed through retrotransposition, horizontal and vertical infection and later by Mendelian heritage [9–12]. Today it is known that 8% of the human genome is composed by sequences of HERVs. HERVs were exposed to several mutation events, and as a result, there are few complete genomic sequences of the classical retroviral genome, since most of their genes are isolated and distributed within the genome, or even many of them are silenced by stop codons within viral genes, which lead to the active replication of HERVs does not occur [13], even though HERVs' protein is still expressed and the virion may be formed [14–17].

HERVs presents a fundamental role in different physiological situations. In this respect, one of the remaining and nearly complete HERV-W env gene is responsible for expressing Syncitin-1, which is a critical protein in the process of Syncytiotrophoblast formation and myogenesis [18–20] In fact, the fusogenic role of HERVs in human myogenesis has been also assessed, and it was described that HERV-W presented a key role in cell fusion of myotubes in athletes engaged in long-duration exercise, such as competitive cycling [21]. Additionally, HERVs

expression in these athletes seems to be diverse, HERV-K was also upregulated in them, which points to the fact the physical exercise may modulate other HERVs expression as well.

Regarding interplay between inflammation and HERV transcription, it was suggested that inflammation might lead to complex and necessary changes to promote HERV expression [22]. To clarify this scenario, previous reports have described that the increase of HERV expression in response to pro-inflammatory cytokines relies on inducing the increase of transcription factor to bind and activate HERVs LTR [23], directly eliciting the increase of HERV pol, gag, and env proteins [24]. Although the interplay of inflammation and HERV expression is most discussed during chronic inflammation situations [25] this interaction in the context of acute inflammation is not fully understood, which can include the response to a session of resistance (strength) exercise training.

Beyond these molecules, the acute inflammation promoted by physical exercise can also elicit microparticle release, which is vesicles originating after stimulus as activation or apoptosis from different cell types (endothelial cells, platelets, monocytes, among others), thus presents distinct functions, not only in the exercise training context but also in the diseases as hypertension [26], hypercholesterolemia [27], systemic sclerosis [28] among others [29].

Based on these pieces of information, it is clear that acute physical exercise is able to regulate and promote distinct patterns of gene expression, which include pathways related to inflammation [29] and muscle repair [30] as well as might modify the circulating levels of cytokines [3] and microparticles [29]. However, until now, the acute effect of a single bout of strength exercise session performed by healthy individuals in the systemic cytokine and microparticle levels is still poorly understood and sometimes contradictory, as well as there is no data regarding the dynamics of HERVs expression in this context. Therefore, in the present study, we aimed to assess the level of HERV-W-env, HERV-K (HML1-10), pro and anti-inflammatory cytokines level concentration and to characterize the circulating microparticles in practitioners and non-practitioners of resistance exercises after an acute strength training session.

## Materials and methods

### Design and population of the study

Healthy adults (n = 47) of both sexes, were recruited and interviewed for collecting information related to their habitual routine of the practice of strength exercise training. The volunteers were oriented to carry out two independent visits to the laboratory on different days. During the first visit, participants responded to a pre-participation questionnaire that included demographic information plus details on their routine of the practice of strength exercise. In accordance with the data obtained in this questionnaire, the volunteers were separated into two groups: regular practitioners of resistance exercise training (REG, n = 27), and non-practitioners of resistance exercise training (control group, CG, n = 20). On the second visit, between 8–10 a.m., a fasting blood sample was collected before the training session (T0). Soon after, a standardized meal was supplied to ensure that all participants had the same amount of calories prior to the strength exercise session. Following 30 minutes, the volunteers performed the exercise session, and 45 minutes after completing this session, the second blood sample was collected (T1). Subjects were instructed not only to maintain their food pattern but also to refrain from alcohol and exercise for 48 hours, as well as caffeine for 24 hours prior to the second visit. This study was performed at the same gym academy and research laboratory, both belonging to the Universidade Santo Amaro (UNISA), São Paulo, Brazil, and all volunteers were students, professors, and employees of the university. Individuals who presented a history of autoimmune diseases, including those in the family, or any inflammatory conditions (such

as diabetes, obesity, neoplasia, or other inflammatory diseases) were excluded from the study. Sampling occurred from 1[th] march 2021 to 1[th] august 2021.

## Ethics

Volunteers were individually informed about the proposed risks and benefits of the study, and those who agreed to participate were oriented read and sign the written consent form previously approved by the Ethics and Research Committee from Santo Amaro University (UNISA, protocol # 4.237.943). It is worth mentioning that the study was performed in agreement with the Declaration of Helsinki, and also with the Ethical Standards presented in 2016 by Harris [31].

## Exercise training protocol

The protocol of the strength exercise session consisted of (i) a general warm-up on a treadmill (E720; Movement®, Pompeia, SP, Brazil) walking at 5 km/h for five minutes followed by 3 minutes of light stretching of the lower and upper limbs; (ii) followed by the performance of strength exercises in three sets of 8–10 maximum repetitions, including ten different types (leg press, high pull, knee extension, elbow flexion, knee flexion, triceps puller, bench press, leg abduction, low pull, leg adduction). It is important to cite that in the occurrence of more or less repetitions (>2) exceeding the target range of 8–10 repetitions in a set, weight correction was applied (1–2%). The recovery interval between sets was 60 seconds. Subjects were oriented to perform a cadence of repetitions with a concentric action of approximately 1 second and an eccentric action of approximately 2 seconds.

## Samples preparation

As previously mentioned, the first blood sample collection occurred in fasting conditions and before the exercise session, whereas the second blood sample collection occurred 45 minutes after the exercise session ended. All blood samples were collected in tubes containing or not anticoagulant EDTA to obtain serum or plasma, as well as Peripheral Mononuclear Blood Cells (PBMC). The PBMC was used to assess the HERV expression, whereas circulating microparticles were analyzed in the plasma and serum samples were used to the cytokine analysis.

## Molecular analysis

The protocol for sample preparation for molecular analysis was conducted as previously described [32–34]. Briefly, total RNA was extracted from PBMCs using the Trizol method (Invitrogen). Rigorous decontamination of genomic DNA was performed with DNA free turbo (Ambion). Absence of contaminant genomic DNA was confirmed by Real Time PCR with primers complementary to GAPDH gene with the absence of Reverse Transcriptase.

## HERV-W and K detection and relative quantification analysis

Around 150 ng of RNA were used to perform one step RT-PCR in a Sybr green assay (Merck). We used primers for env-HERV-W [35], gag-HERV-K (HML1-10) [32], and GAPDH as housekeeping gene [36]. The RT-PCR mix included 0.1 μM of each primer and 1x of PCR Master Mix Sybr-Green one step (Merck). For both HERV-W and HERV-K cycling conditions were 50˚C for 2 minutes, 95˚C for 10 minutes followed by 40 cycles, of 95˚C for 1 minute, 50˚C for 45 seconds and 60˚C for 1 minute. For GAPDH assay the cycling conditions were: 50˚C for 2 minutes, 95˚C for 10 minutes followed by 40 cycles of 95˚C for 1 minute and 60˚C for 1 minute. In both assays a previous step was added of 37˚C for 30 minutes for cDNA synthesis and a final cycle to determine the melting temperature of the samples (55˚C to

95°C). HERV activity expression was qualitatively (absence/presence) and quantitatively (level of expression) evaluated. The level of expression was determined by the $2^{-\Delta\Delta Ct}$ method where $\Delta Ct$ = (HERV Ct- GAPDH Endogenous Control Ct)–(Average of $\Delta Ct$ of all controls), and the results were represented as fold changes. Significance was evaluated using Wilcoxon test. In all cases, samples were only considered positive for HERVs if melting curve was the same or -+0.3°C distinct of the control samples, and therefore included in the relative quantification analysis.

## Determination of systemic cytokine concentration

Serum concentrations of the pro-inflammatory IL-6 and TNF-α cytokines and the anti-inflammatory IL-10 cytokine were determined by using the ELISA commercial kits (ThermoFisher) following manufacturer's instructions. The concentration of each cytokine was calculated through an appropriate standard curve that presented a correlation coefficient from 0.95 to 0.99, with coefficients of variance intra-assay varying from 2,5 to 4% and from 8 to 10% in inter-assay.

## Quantification and phenotypic characterization of circulating microparticles

Samples were centrifuged at 160 g, 22C for 10 minutes to obtain platelet-rich plasma (PRP). The PRP was then centrifuged at 1500g, 22C for 6 minutes, to collect platelet-poor plasma (PPP). Then, the PPP (70 μL) was incubated for 20 minutes at room temperature with anti-CD105 conjugated with APC, anti-CD42 conjugated with FITC combined with anti-CD31 conjugated with PE and anti-CD14, to identify endothelial, platelet and monocytic microparticles, respectively; isotypes were used as controls (BD Biosciences, Franklin Lakes, NJ, USA). After incubation, 300 μL of PBS were added and immediately read in a flow cytometer (FACS-Calibur—BD Biosciences, Franklin Lakes, NJ, USA), with approximately 30,000 events were acquired.

## Statistical analysis

For data analysis, SPSS version 26.0 was used. The adherence to Gaussian distribution and homogeneity of variance were evaluated using Kolmogorov-Smirnov and Levene tests, respectively. The data are presented as mean ± standard deviation or median (interquartile range), as appropriate. Paired Student-T test, or Wilcoxon test were used for comparisons between the time points (before and after exercise session). To compare the two groups, we used the Kruskal-Wallis test with Müller–Dunn posthoc test. The significance level was set at $p < 0.05$. Statistic difference was assumed if $p < 0.05$.

Spearman's rank correlation coefficient test was applied to verify the occurrence of association among the parameters assessed here.

Sample size was conveniently collected. To compare data obtained between the groups, parametric and non-parametric tests were used.

## Results

The REG performed in their habitual routine around 4.2±1.5 sessions of strength exercise training per week, with 9±4 types of exercises per session, 3±1 sets, and 12±3 repetitions. The most common types of exercises were bench press, barbell curl, bull down, pull-up, row down, leg press, leg curl, leg extension, and squat. The demographic findings are described in Table 1.

**Table 1. Demographic data of the individuals who attended to the study.**

| Groups | N | Sex | | Age, M+SD | BMI+SD |
|---|---|---|---|---|---|
| | | female | Male | | |
| CG | 20 | 10 | 10 | 25.4±8.9 | 24.65±4.1 |
| TG | 27 | 14 | 15 | 25.1±5.8 | 25.04±3 |
| Total | 47 | 24 | 25 | 25.2±7.4 | 24.87±3.9 |

CG: Control Group, TG: Trained Group. M+SD: Median and standard deviation, BMI: Body Mass Index

Fig 1 shows the level of HERV-W (Fig 1A) and HERV K (Fig 1B) expression in the volunteer groups, both before (T0) and after (T1) the strength exercise session. It was possible to evidence that all volunteers presented some level of HERV K and W expression at T0, and also that no significant discrepancy was observed in the level of expression between women and men (p = 0.780). Particularly in terms of the HERV-K (HML1-10) expression, neither statistically significant differences were found between the volunteer groups, nor in a comparison between the time points within groups (Fig 1B). However, HERV-W expression significantly increased (2-fold) from T0 to T1 in REG (p<0.01), whereas CG showed that the expression profile was unchanged (p = 0.8, Fig 2B). In agreement with data presented in Fig 2, in an interesting way, 19 out of 20 individuals in REG upregulated HERV-W following the exercise session (T1), whereas only 6 individuals (30%) from CG upregulated this HERV, and the remaining 14 volunteers (70%), actually, downregulated its expression (2.7-fold average) at T1.

Although most individuals presented HERV-K expression at some level, we did not find statistical difference regarding HERV-K (HML1-10) expression, neither between groups nor comparing the time points within groups (Fig 3).

Beyond the HERV K and W expressions, in Fig 3 is shown the results obtained in the assessment of the systemic concentration of both pro- and anti-inflammatory cytokines before (T0)

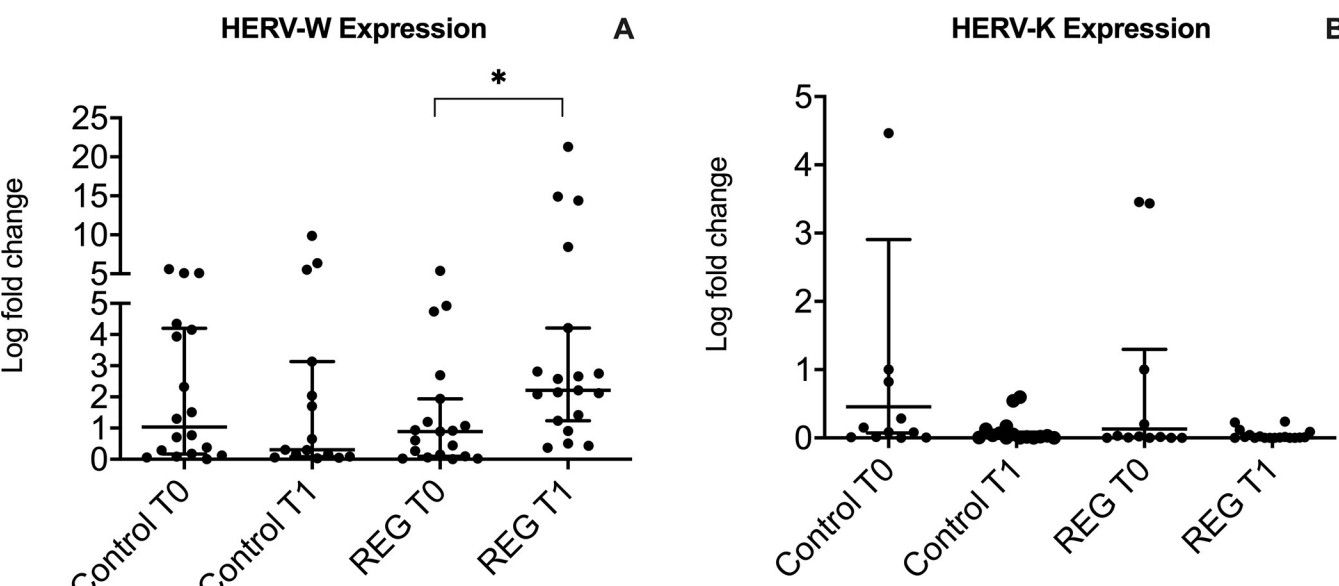

**Fig 1.** Profile of HERV-W (A) and K (B) expression before and after the training session. Expression levels are shown in fold change. * = p<0.01 Wilcoxon test, T0: sampling before the training session, T1: sampling 45 minutes after the training session. REG = regular practitioners of resistance exercise training group.

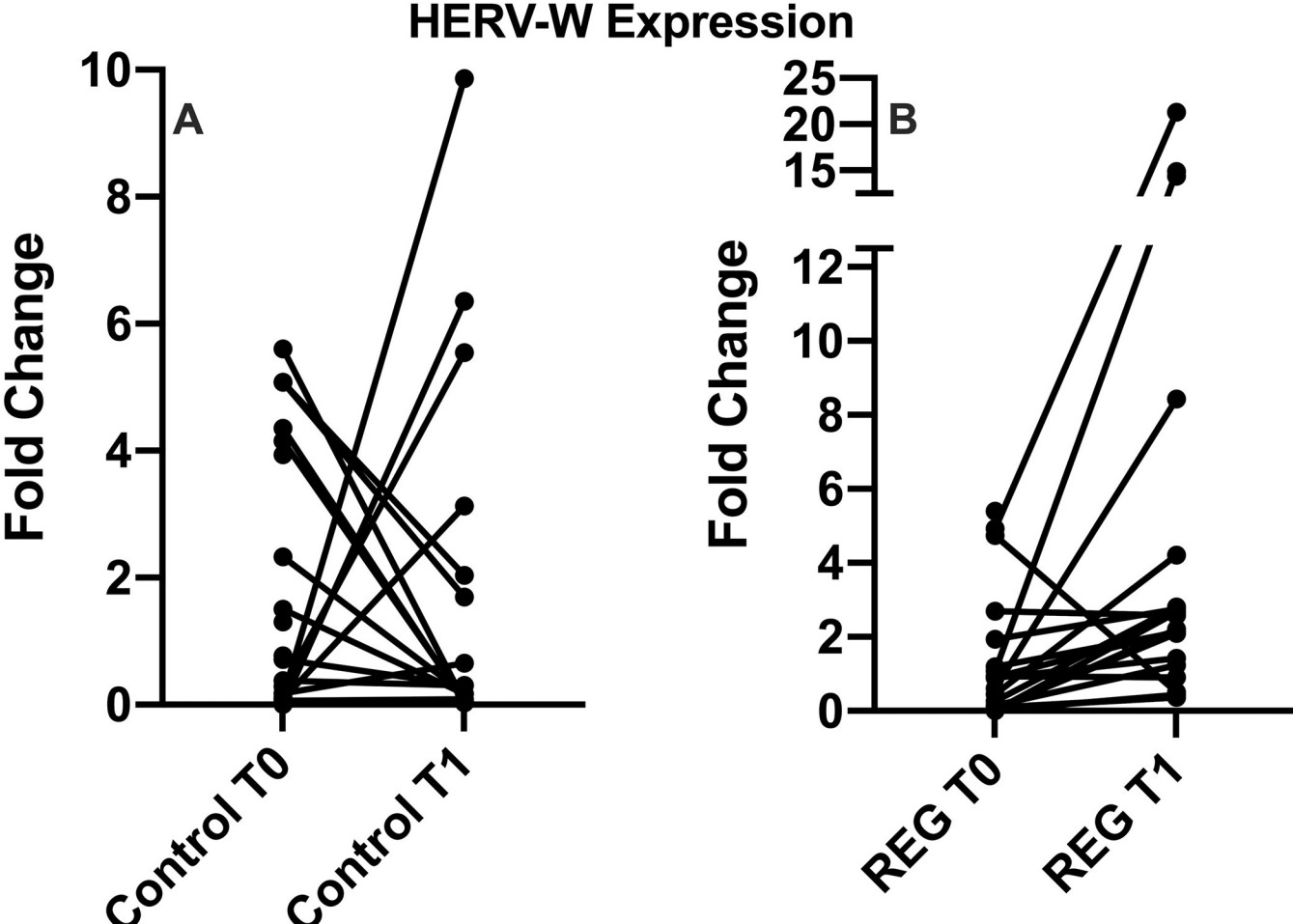

**Fig 2. Profile of HERV-W expression before and after the training session.** Lines show paired individuals. T0: sampling before the training session, T1: sampling 45 minutes after the training session.

and after the exercise session (T1). Higher serum levels of IL-10 (Fig 3A) and TNF-α (Fig 3B) were found at T1 than at T0 in both volunteer groups (REG, p<0.0001 for IL-10 and p<0.0001 for TNF-α; CG, p = 0.05 for IL-10 and p<0.01 for TNF-α). In addition, the serum IL-6 levels were not different between the groups and time points assessed here (Fig 3C). We had performed correlation analysis between the concentration of cytokines variables and the HERVs expression, and a positive correlation was observed in REG between the serum levels of IL-10 at T0 and TNF-α at T1 (p = 0.02, Fig 3D).

Finally, Table 2 shows the results obtained in the quantification and characterization (POR of endothelial (EMP), platelet (PMP), and monocytic microparticles (MMP) in the platelet-poor plasma from the volunteer groups both before (T0) and 45 minutes after the strength exercise session ended (T1). It was possible to observe that REG showed a significant decrease in EMP at T1 (p = 0.02) as compared to the values at T0. No other differences were found.

## Discussion

In the present study, for the first time, we were able to show not only an acute and significant increase in HERV W expression, but not of HERV K, as well as reduction in the endothelial

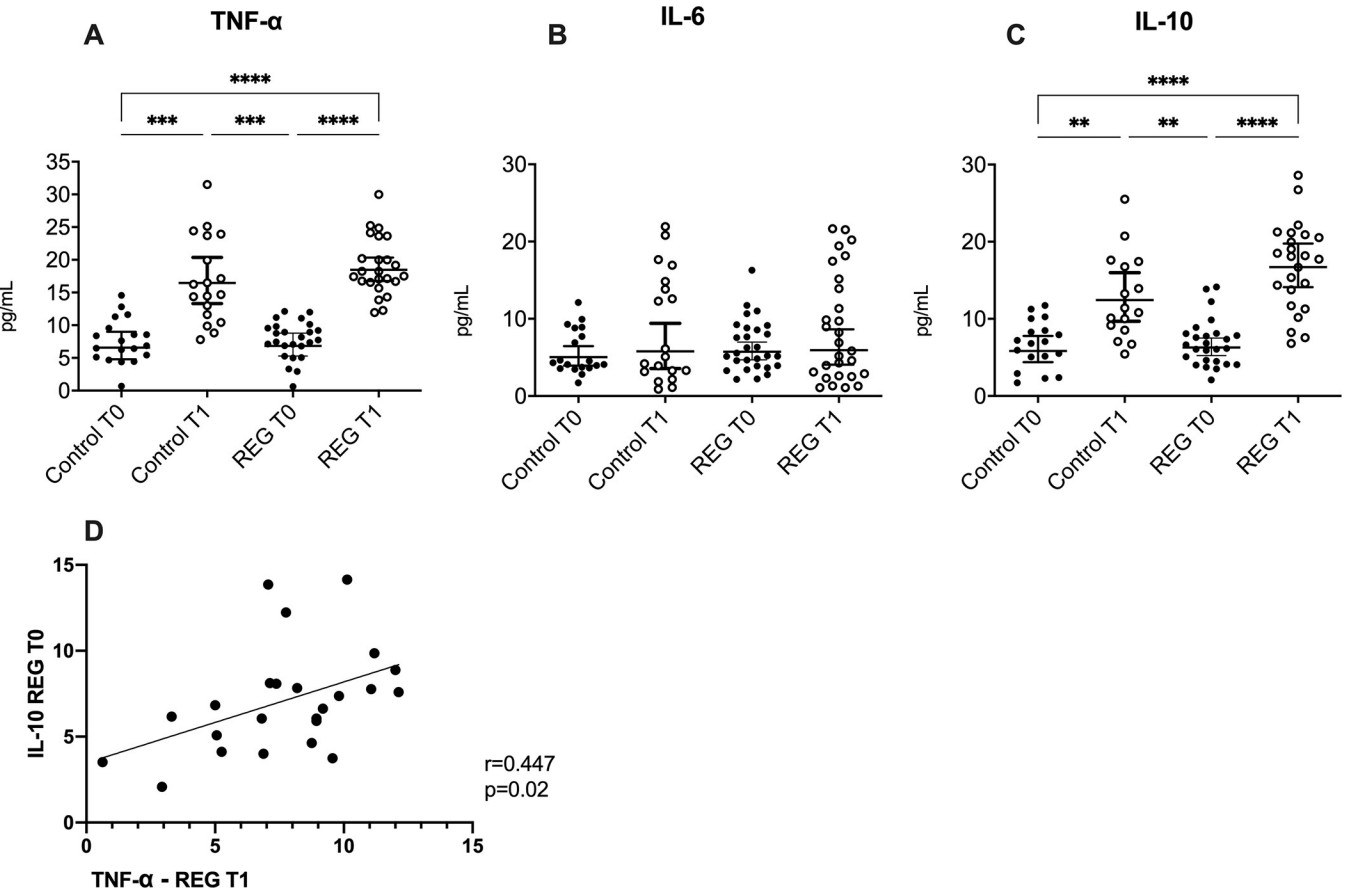

**Fig 3.** Serum concentrations of TNF-α (A), IL-6 (B), and IL-10 (C), both before (T0) and 45 minutes after the strength exercise session ended (T1) in the practitioners of resistance exercise training group (REG) and control group (CG). In addition, Pearson's coefficient correlation analysis between values of IL-10 at T0 and TNF-α at T1(D). **:p<0.05 ***: p<0.01 ****: p<0.0001.

microparticles (EMP), in response to a strength exercise session in a group of individuals who regularly practiced resistance exercises. Moreover, as expected, the circulating levels of some cytokines, particularly TNF-a and IL-10, also increased after the exercise session in both volunteer groups.

HERVs have been implicated in a range of roles within human physiology and disease, as documented previously [18, 37, 38]. Specifically, due to its fusogenic activity Syncytin-1 has been directly linked in the formation of osteoclasts and myoblasts [37, 39]. In this study, we report an increase in HERV-W expression in individuals that exercises regularly, following an acute strength training session but not in the control group.

**Table 2. Percentage of circulating microparticles in platelet-poor plasma from a group of practitioners of resistance exercise training (REG) and non-practitioners (control group—CG) both before (T0) and 45 minutes after strength exercise session ended (T1).** The data is represented as medians (interquartile range). Wilcoxon test was used in intragroup comparisons (T0 x T1 in each group) and Mann-Whitney test was used in the comparisons between RGE and CG, for each visit.

| Microparticles | GC | | p intra groups | REG | | p intra groups | p between groups | |
|---|---|---|---|---|---|---|---|---|
| | T0 | T1 | | T0 | T1 | | T0 | T1 |
| **Endothelial** | 0.06 (0.03–0.21) | 0.02 (0.00–0.18) | 0.23 | 0.03 (0.01–0.11) | 0.01 (0.00–0.04) | **0.02** | **0.03** | 0.47 |
| **Platelet** | 73.01 (49.63–86.03) | 78.07 (56.66–90.42) | 0.36 | 70.69 (42.02–85.93) | 75.16 (33.01–85.64) | 0.89 | 0.81 | 0.42 |
| **Monocytic** | 2.09 (0.80–5.81) | 1.79 (0.13–4.20) | 0.14 | 1.66 (0.89–4.23) | 2.93 (0.15–8.29) | 0.10 | 0.65 | 0.19 |

These findings underscore the rapid expression of HERV-W immediately following an acute training session in physically trained individuals. This phenomenon may suggest an acquired adaptation of the organism to the expression of this retroelement, potentially signifying a vital physiological role. However, a more comprehensive understanding of this role requires further investigation.

Several plausible explanations may account for these findings. First, Syncytin-1 could be directly associated in the muscle repair process. In fact, it has been documented that Syncitin-1 can interact with other sarcolemma proteins, such as caveolin, which plays a role in muscle repair [39–42]. Thus, individuals who are well-adapted to physical exercise might exhibit a positive regulation of HERV-W expression due to their enhanced physical conditioning. However, it is important to note that our study is based on peripheral blood mononuclear cells (PBMCs), and the direct impact of Syncytin-1 on muscle repair warrants further investigation, and in this study our primer set was not directed specifically to Syncytin-1. Secondly, the higher and significant increase in the HERV-W expression in REG individuals may be explained to the gene modulation pattern that is acquired due to the regular training [43], and therefore HERV-W expression may also be regulated by training. Although this has not been observed for HERV-W previously, the distinct profile of gene expression on chronic and acute exercise is not new. Previous studies focused on understanding the gene expression and micro-RNA during physical exercises showed distinct patterns of gene expression in acute and chronic exercise [44, 45].

Interestingly, a limited number of studies have investigated the frequency and level of HERVs expression in the context on physical exercise, as well as their involvement in the muscle repair process. A prior study described that prolonged endurance exercise was able to modulate the expression HERV-W in the skeletal muscle tissue, thereby promoting tissue repair [21]. This finding is also in touch with ours, since HERV-W was upregulated in trained individuals, and specially in acute session of training and in strength exercise profile, differently than the observed in the untrained individuals. Therefore, we might envisage a scenario where HERV-W expression may participate to promote muscle repair and this high expression in trained individuals might be an acquired condition that is determined by the frequency of training. In the opposite side, HERV-W expression was downregulated (2.7 fold change) in the individuals who do not perform regular strength exercise. This distinct pattern between groups might suggest the HERV-W expression behave differently accordingly the physical exercise routine and might be associated to the repair condition.

Interestingly, HERV-K (HML-1) was not differentially expressed in REG patients and CG, this is interestingly, since HERV-K is the newest and most active family. Therefore these findings are critical since the expression of one family but not others is not occurring as a consequence of epigenetic changes after cellular stress, caused by local inflammation, but rather, the HERV regulation may be intricately linked to a very specific role in muscular repair. And this make sense since syncytin-1 is responsible for fusogenic events commonly observed in different cell types formation, such as syncytiotrophoblast [11] and osteoclasts [14, 15].

Beyond these aspects, it is worth mentioning that, in agreement with the literature, not only an acute exercise session is able to elicit several gene transcriptions but also it was reported that after 45 minutes more than 50% of these genes can be upregulated [46]. Specifically, resistance training can activate important signaling pathways 45 minutes after a bout of this exercise type as compared with rest. Moreover, it was reported that, at this time point, both restoration of tissue homeostasis and integrity, and also protein synthesis and degradation are still being regulated [47], and, interestingly, the human skeletal muscle is more sensitive to acute resistance training in the trained than untrained individuals, which reinforce the adaptive process that occurs in response to repeated stimulation [48]. These pieces of information

may help us to understand the great changes that can be observed between 45 to 60 minutes after a bout of exercise in some molecules released in response to muscle contracting, such as TNF-α, IL-6 and IL-10 [49].

In an interesting way, previous finding reveals that inflammation may interfere in the HERV expression [50, 51], and, here, in both groups was found higher TNF-α levels, a pro-inflammatory cytokine, however the level of HERV-W expression in the CG was unchanged between T0 and T1, which reinforces a possible acquired modulation of HERV-W expression to be associated to skeletal muscle repair.

Regarding the literature, whether one side, not only the HERV-W family can elicit the production of some proinflammatory cytokines, e.g. IL-1β, IL-6, and TNF-α, by its interaction with Toll-like receptors (TLRs), particularly TLR4 and CD14, which shows that HERV trans-activation can act fueling the inflammation, as well as HERVs can activate the immune response, especially innate immunity, driving an uncontrolled inflammation that can towards of chronic inflammation [52], or even some HERV Env proteins, particularly derived by HERV-H, HERV-K, and HERV-FRD (Syncytin-2), could be involved in immunosuppressive aspects [53], by induction of the anti-inflammatory cytokine IL-10 [54, 55], on another side, it was demonstrated that TNF-α is capable of elevating the RNA expression of HERV-H, HERV-K, and HERV-W [23] through TNF-α receptor signaling. Taken together, these data show a close interplay between HERVs and inflammation.

Based on these pieces of information, besides assessing HERV K and W expression, we also evaluated the circulating levels of some cytokines, such as IL-6, IL-10, and TNF-α, which have presented remarkable interplay within the physical exercise context [56]. Since the last decades from the 20th century, the number of reports showing that physical exercises can elicit prominent alterations in systemic cytokine levels gradually increased. In this sense, nowadays it is broadly known that the muscle contractions in response to physical exercise performance induce the transcription of several genes, which include different types of cytokines [3]. Specifically in this context, the cytokines released from contracting skeletal muscles, named myokines, can exert autocrine, paracrine, and endocrine actions in many organs and tissues, which include lipolysis, glycolysis, and pro- and anti-inflammatory effects [3, 57]. In fact, the first and most studied myokine is IL-6 and it was demonstrated that their circulating levels can rapidly increase (up to 100-fold) in response to a physical exercise session [57] dropping in minutes/hours post-exercise [3]. In fact, it was suggested that the myokine IL-6 can work as an energy sensor [58], especially in musculoskeletal tissue, since was documented that musculoskeletal tissue from exercising leg with reduced glycogen content also presented increased IL-6 mRNA levels. Moreover, it is worth mentioning that the same authors cited that the IL-6 release from the exercise leg, in this context, occurred after one hour of exercise [59, 60]. Taken together, these pieces of information can corroborate our lack of differences in the circulating IL-6 levels, both between the groups and time points assessed here, since all volunteers received the same standard meal before the strength exercise training session, which could be provided sufficient "fuel" to perform the exercise session and the musculoskeletal glycogen was not significantly reduced. Furthermore, we can also suggest that, since the second blood sampling occurred 45 minutes after the exercise session ended, the IL-6 release from musculoskeletal tissue probably was not sufficient, until this moment, to significantly alter the circulating IL-6 levels. Corroborating this last suggestion, it was suggested that eccentric exercise does not necessarily elicit a larger increment of plasmatic IL-6 levels than concentric "nondamaging" muscle contractions, which not only shows that the elevation of systemic IL-6 is not closely associated with muscle damage but also eccentric exercise can lead to a delayed peak, as well as a slow dropped of this myokine in the recovery period [61].

Despite the circulating IL-6 levels were not altered, we verified that the systemic levels of TNF-α and IL-10 significantly increased after the strength exercise session ended in both volunteer groups. It is well-accepted that during an acute-phase response, the local inflammatory response induced usually is followed by a systemic response and that a network of several cytokines, with pro and anti-inflammatory properties, present corollary actions [56]. Specifically in the physical exercise context, it was demonstrated that TNF-α levels (both the mRNA in skeletal muscle and protein in the circulation) are, acutely, most prominently increased after the eccentric exercise [56, 62, 63] than after concentric exercise [64]. It was proposed that this proinflammatory myokine could be an important player in the initiation of the breakdown of damaged muscle tissue [65, 66]. In controlled situations, which include both production and regulation, TNF-α can assist in the muscle regeneration response following injury, and, after binding to its TNFR1 receptor, several pleiotropic signaling pathways are activated. In particular, it was reported that TNF-α not only prevents myoblast fusion, perhaps by its modulation of both caveolae organization/function and satellite cell function, but also TNF-α can improve the myoblast's resistance to osmotic stress. Thus, these pieces of information show a prominent modulatory effect of TNF-α in both fusion and stress resistance of plasma membranes of muscle cells [67], which can putatively help the muscle regeneration of muscle skeletal tissue after the injury promoted by the strength exercise session.

Although there is a consensus that the activation of an inflammatory response is necessary to promote muscular regeneration, it is also well-known this process needs to be controlled. In fact, it was documented that, in a similar way to pro-inflammatory cytokine response, after the physical exercise session there is an increase in circulating levels of anti-inflammatory cytokines, such as the IL-10, both to prevent excessive secretion or even control the effect of pro-inflammatory cytokines, mainly TNF-α [56, 68]. Concerning resistance/strength exercise training, it was reported that this type of exercise can increase the IL-10 mRNA expression in muscle tissue, besides an increase of circulating IL-10 levels in conjunction with reduced systemic TNF-α concentrations [69]. Therefore, these pieces of information corroborate our findings not only in that the elevation of circulating IL-10 levels after the exercise session ended shows that the necessary mechanism of inflammation control was present in both volunteer groups in order to avoid a possible dangerous effect related to an exacerbation of pro-inflammatory cytokines, mainly TNF-α, but also can suggest that regular practice of resistance exercise training is able to improve the inflammatory response to this physical exercise since it was evidenced that the volunteers in REG showed a significant positive correlation between the circulation levels of IL-10 at T0 with TNF-α at T1.

Our results showed a decrease in EMP in the REG after training. In fact, the studies are inconclusive concerning the effect of exercise in the microparticle levels, since the type, intensity, duration, and frequency showed different results. Furthermore, concerning EMP, there are many protein surfaces described in the literature for recognizing these microparticles (CD51, CD105, CD144, CD31, among others), which becomes difficult to compare the different findings.

Bittencourt et al. evaluated the chronic effect of the exercise in professional runners, compared to sedentary individuals, and did not find differences in the percentage of EMP in the runners, although there was higher levels of endothelial progenitor cells (considered biomarkers of vascular repair), suggesting that the prolonged exercise promotes benefits on vascular homeostasis [70]. Another study showed reduced levels of EMP after a moderate intensity training [71], a reduction in the EMP levels immediately after a single bout of exercise was also reported [72], pointing out to an improvement in endothelial function promoted by the exercise, even a single training, which are in line with our findings.

In summary, we have described distinct pattern of HERV-W expression regularly exercised individuals and non-trained individuals, and this might be indicative for a possible role in muscle repair. Although this endogenous retrovirus has been associated to many diseases, especially autoimmune conditions, clear evidences describe the participation of HERVs on many physiological activities, including myogenesis. Importantly, the data described here can not only improve our understanding of the acute impact of a single bout of a strength exercise session on these retroelements but also can indicate that HERV is an important target involved in this context and that it might putatively be involved in the muscle repair process. In this sense, further studies should focus on possible pathways analysis in order to precisely determine HERV-W role and other retroelements in muscle repair.

The study presents some limitations, as we have analyzed only the expression of HERV-W, and we did not focus on the detection of protein itself or Syncitin-1 specific expression, although HERV-W. However, we still lack data regarding the modulation of HERV-W expression in physical exercise. The second limitation pertains to the use of PBMC as the preferred sample for assessing gene expression. While studying the skeletal muscle tissue itself would be valuable for detecting local protein/ gene expression, several studies have demonstrated that PBMC serves a suitable model for studying HERV expression [32, 73–75]. Finally, our study focused on a single training session, and consequently, we did not follow up with these individuals to understand the overall dynamics of HERVs activity, circulating microparticles and cytokines. To address this, further studies should explore the expression and protein detection in both skeletal muscle tissue and PBMC and in different conditions of training and our findings revealed that this study brought new findings that might add pieces to the puzzle to understand whether HERV-W may play a role in skeletal muscle repair.

## Conclusions

Higher level of HERV-W expression was found in REG individuals and important increase was found in most of the REG individuals after training session, whereas CG most of individuals have shown a decrease in the level of HERV-W expression. TNF-$\alpha$ and IL-10 was increased in REG and CG after training, and lower concentration of EMP was found in REG individuals.

## Supporting information

**S1 Raw data.**
(XLSX)

## Author Contributions

**Conceptualization:** Lucas Vinicius Morais, Camila Malta Romano, Marina Tiemi Shio, Lucas Melo Neves, Luiz Henrique da Silva Nali.

**Data curation:** Lucas Vinicius Morais, Samuel Nascimento dos Santos, Camila Malta Romano, Patricia Colombo-Souza, Jonatas Bussador Amaral, Marina Tiemi Shio, Lucas Melo Neves, André Luis Lacerda Bachi, Carolina Nunes França, Luiz Henrique da Silva Nali.

**Formal analysis:** Lucas Vinicius Morais, Samuel Nascimento dos Santos, Tabatah Hellen Gomes, Jonatas Bussador Amaral, André Luis Lacerda Bachi.

**Funding acquisition:** Marina Tiemi Shio, André Luis Lacerda Bachi, Luiz Henrique da Silva Nali.

**Investigation:** Lucas Vinicius Morais, Tabatah Hellen Gomes, Marina Tiemi Shio, Luiz Henrique da Silva Nali.

**Methodology:** Lucas Vinicius Morais, Samuel Nascimento dos Santos, Tabatah Hellen Gomes, Patricia Colombo-Souza, Jonatas Bussador Amaral, Carolina Nunes França.

**Project administration:** Lucas Vinicius Morais, Patricia Colombo-Souza, Luiz Henrique da Silva Nali.

**Resources:** Tabatah Hellen Gomes, Patricia Colombo-Souza, Jonatas Bussador Amaral.

**Software:** Jonatas Bussador Amaral.

**Supervision:** Lucas Melo Neves, Luiz Henrique da Silva Nali.

**Validation:** Samuel Nascimento dos Santos, Patricia Colombo-Souza, Lucas Melo Neves, Luiz Henrique da Silva Nali.

**Visualization:** Lucas Vinicius Morais, Samuel Nascimento dos Santos, Carolina Nunes França, Luiz Henrique da Silva Nali.

**Writing – original draft:** Lucas Vinicius Morais, Camila Malta Romano, Lucas Melo Neves, Carolina Nunes França, Luiz Henrique da Silva Nali.

**Writing – review & editing:** Camila Malta Romano, Marina Tiemi Shio, Lucas Melo Neves, André Luis Lacerda Bachi, Carolina Nunes França, Luiz Henrique da Silva Nali.

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
