## [Decision Letter · Decision Letter 0]

6 Mar 2024

PONE-D-24-05279Acute strength exercise training impacts differently the HERV-W expression and inflammatory biomarkers in resistance exercise training individualsPLOS ONE

Dear Dr. Nali,

Thank you for submitting your manuscript to PLOS ONE. After careful consideration, we feel that it has merit but does not fully meet PLOS ONE’s publication criteria as it currently stands. Therefore, we invite you to submit a revised version of the manuscript that addresses the points raised during the review process.

We look forward to receiving your revised manuscript.

Kind regards,

Matthew Cserhati, Ph.D

Academic Editor

PLOS ONE

Journal Requirements:

   "Fundação de Amparo à Pesquisa do Estado de São Paulo (FAPESP), grants # 2023/08773-0 and 2020/11619-5"

4. We note that your Data Availability Statement is currently as follows: All relevant data are within the manuscript and its Supporting Information files

Additional Editor Comments:

Acute strength exercise training impacts differently the HERV-W expression and inflammatory biomarkers in resistance exercise training individuals

P ONE-D-24-05279

Smaller comments:

1. Line 54: “Inflammation is a vital and evolutionary conserved process…” Evolution means change, and conservation means stasis, so this term–though widespread–is an oxymoron. Use a better term, something like “ubiquitous”.

2. 58: well-coordinate: do you mean well-coordinated?

3. 65: do you mean (IL-6) ?

4. 73: that rises promptly …

5. 83: the transcription of

6. 87: Mendelian…

7. 304: what does p=0.0XX mean? Did you want to write a number in place of the X?

Specific, detailed comments:

1. Someone should read the paper and correct the English, there are slight mistakes here and there.

2. 82-89: The idea that HERV’s are ancient viruses that infected the human genome millions of years ago is contested. There is no direct evidence for this. For example, some researchers have studied the Arc gene, which modulates synaptic plasticity in the brain. The Arc mRNA is able to assemble itself into virus-like capsids that encase RNA, exhibiting traits of retroviral Gag proteins. See Pastuzyn et al., 2018., Cell. These HERV’s instead might be design structures in the genome. This being said, is it not possible that HERV-W and -K are actually integral parts of the human genome that are activated during physical exercise?

3. 306-7: there was a positive correlation between IL-10 (T0) and TNFa (T1), but the correlation is moderate at best. Also, why did you compare two different genes at two different time points? Why not IL-10 (T0) versus TNFa (T0) and IL-10 (T1) versus TNFa (T1)?

4. 338: Why is it significant that HERV-W is induced, but not HERV-K?

5. 480-88: The present study presents interesting information, but adding protein analysis would strengthen your results. Also, if possible, it would be even better if you could follow up with the study participants. Hence, I would like to return your paper for major revisions.

Reviewers' comments:

Reviewer's Responses to Questions

**Comments to the Author**

1. Is the manuscript technically sound, and do the data support the conclusions?

Reviewer #1: Yes

2. Has the statistical analysis been performed appropriately and rigorously? 

Reviewer #1: Yes

3. Have the authors made all data underlying the findings in their manuscript fully available?

Reviewer #1: Yes

4. Is the manuscript presented in an intelligible fashion and written in standard English?

Reviewer #1: Yes

5. Review Comments to the Author

Reviewer #1: Dear authors to this manuscript draft,

I understand that the HERVs expression and inflammation profile may be the one of the factor to repair muscles in practitioners’ resistance exercises after an acute strength training session. Your explanations are very concise and easy to understand. I would like to suggest the following four points to express your hypothetical proof more clearly.

Major points

I think it would be better to use induction to explain how the expression of HERV-W is regulated by cytokines since you are saying that muscle repair is promoted by changes in the cytokine HERV-W, it would be better to explain what has been revealed about the relationship between cytokines and HERV-W.

It would be helpful to provide the rationale for why the time point was set at 45 minutes. I think signals of muscle repair from exercise should be visible for several days, but it would be helpful to provide evidence that it is best to see them after 45 minutes with your chosen training method.

I think it would be better to clearly state whether the exercises in this method are eccentric or concentric, if possible. This is because I think there is a distinction between eccentric and concentric movements when discussing their effects.

Syncytin-1 is mentioned in the introduction and discussion, but if it is important for the hypothesis of MREV-W expression and muscle repair in this experiment, it would be better to show data on the expression of syncytin-1 as well. think. That will be more persuasive.

Minor points

I thought it would be better to change the discrepancy on line 276 to the word difference.

6. PLOS authors have the option to publish the peer review history of their article (what does this mean?). If published, this will include your full peer review and any attached files.

Reviewer #1: No

---

## [Author Response · Author response to Decision Letter 0]

21 Mar 2024

March, 12th 2024 

Professor Mathew Matthew Cserhati, Ph.D

Academic Editor, Plos ONE

Dear Professor,

We are pleased for the evaluation of our paper by the reviewer and the editor. In this letter we bring answers to all the questions raised by them. We are also submitting the reviewed version of the manuscript for your consideration.

Editor comments:

Authors’ Answer: We have now uploaded the updated versions of manuscript files according to the provided templates

Authors’ Answer: We have now removed this information of the manuscript as requested.

 "Fundação de Amparo à Pesquisa do Estado de São Paulo (FAPESP), grants # 2023/08773-0 and 2020/11619-5"

Authors’ Answer: The funders had no role in study design, data collection and analysis, decision to publish, or preparation of the manuscript. We have also included this statement in the updated cover letter

4. We note that your Data Availability Statement is currently as follows: All relevant data are within the manuscript and its Supporting Information files

Please confirm at this time whether or not your submission contains all raw data required to replicate the results of your study. Authors must share the “minimal data set” for their submission. PLOS defines the minimal data set to consist of the data required to replicate all study findings reported in the article, as well as related metadata and methods

Authors’ Answer: We are submitting at this time all the raw data required to replicate the results of the study. 

Additional Editor Comments:

Acute strength exercise training impacts differently the HERV-W expression and inflammatory biomarkers in resistance exercise training individuals

P ONE-D-24-05279

Smaller comments:

1. Line 54: “Inflammation is a vital and evolutionary conserved process…” Evolution means change, and conservation means stasis, so this term–though widespread–is an oxymoron. Use a better term, something like “ubiquitous”.

Authors’ Answer: We thank the editor for the term suggestion, it was changed as suggested. 

2. 58: well-coordinate: do you mean well-coordinated?

Authors’ Answer: Yes, we do. The sentence was corrected accordingly. Thank you.

3. 65: do you mean (IL-6) ?

Authors’ Answer: Yes, it was a typo and we apologize for that.

4. 73: that rises promptly …

5. 83: the transcription of

6. 87: Mendelian…

Authors’ Answer: We thank for the editor revision, and we apologize for the grammar mistakes, we have thoroughly reviewed the text for any more typos and grammar errors. 

7. 304: what does p=0.0XX mean? Did you want to write a number in place of the X?

Authors Answer: We did. In fact, this information was missing in the text, we have added it. 

1. Someone should read the paper and correct the English, there are slight mistakes here and there.

Authors’ Answer: We thank the editor for point this out. We have thoroughly reviewed the text for any remaining typos and grammar errors. Additionally, we have sent this version to a native English speaker for comprehensive review. 

2. 82-89: The idea that HERV’s are ancient viruses that infected the human genome millions of years ago is contested. There is no direct evidence for this. For example, some researchers have studied the Arc gene, which modulates synaptic plasticity in the brain. The Arc mRNA is able to assemble itself into virus-like capsids that encase RNA, exhibiting traits of retroviral Gag proteins. See Pastuzyn et al., 2018., Cell. These HERV’s instead might be design structures in the genome. This being said, is it not possible that HERV-W and -K are actually integral parts of the human genome that are activated during physical exercise?

Authors’ Answer: We thank the editor for this consideration and for the opportunity to clarify this interesting hypothesis. 

We would like to answer the second question first: “…is it not possible that HERV-W and -K are actually integral parts of the human genome that are activated during physical exercise?” Indeed, HERV-K and HERV-W are integral parts of the human genome, along with other endogenous retroviral families, collectively constituting approximately 8% of the human genome. Except for a few polymorphic proviruses from K family, all human being has the same HERV composition, which allow us to say that ERVs make us humans. So the Editor is right, the HERV-W is working as a human (though domesticated) gene that is reactivated under certain conditions.

 The origin of these retroviral sequences is object of study of many researchers. While the idea of ancestral retroviral endogenization events occurring millions of years ago seems intriguingly, previous studies, using comparative genomics and evolutionary approaches were able to estimate time of these insertions into our ancestral’s genome. In the same line of investigation, several papers elegantly described the dynamics of the distribution of these ERVs among different species[1–6]. Interestingly, it was also observed that during host speciation events, the gain/loss and fixation of the proviruses were dramatically impacted by demographic conditions [7]. Using orthologs elements, a team from Imperial College London demonstrated that a particular member of ERV-L, one of the most ancient integrated families, predates the divergence of the placental mammals, and conserved fragments are present also in primates[8]. But not all ERVs are that old. Some families are specific from primates, as HERV-W and the newest one, HERV-K, present only in Old World primates since they integrated in the ancestral genome after the divergence of the New and Old World primates. Still regarding HERV-K, some of them, such as HERV-K-113 and HERV-K 115 frequency vary depending on the population, being present in up to 23% and 35% of the population[9]. 

Maybe the best example, in real time, of the endogenization process are the koala retroviruses. Koala Retrovirus (KoRV) is an important exogenous retrovirus that represent a threat to Koalas. This virus can be either found in genome of some Koala’s population as an endogenized element, verified by high proviral load, or in the exogenous and infectious condition. These conditions are found in particular areas in the island of northeastern coast of Australia[10]. Phylogenetic analysis of KoRV-A LTRs revealed that this integration event had begun around 50.000 years ago and is still in course [11]. This piece of evidence is clarifying how this complex event is occurring in a specific condition. 

Finally, to our understanding, Patuzyn et al., 2018 does not contest HERVs as ancient retroviral integrations but rather describes that Arc possesses molecular properties resembling those of retroviral Gag proteins. In their work, they demonstrated that Arc is derived from a vertebrate lineage of Ty3/gypsy, which is a retrotransposon (considered ancestors of modern retroviruses), not a retrovirus. And, as is the case with all vertebrate genomes, there are several mobile elements from different classes (ranging from DNA transposons to RNA long and short elements such as LINEs and SINEs) as well as retrotransposons and retroviruses. HERVs are integral part of the genome and as described in lines 74-79 (updated version), events of gene domestication [12] occurred throughout the evolution and are crucial gene regulation[13,14] and also for the human physiology.

3. 306-7: there was a positive correlation between IL-10 (T0) and TNFa (T1), but the correlation is moderate at best. Also, why did you compare two different genes at two different time points? Why not IL-10 (T0) versus TNFa (T0) and IL-10 (T1) versus TNFa (T1)?

Authors Answer: We thank the editor for this consideration. Actually, we have performed the correlation analysis in all possible combinations, including the ones highlighted by the editor. However we did not observe any significant positive or negative correlation between these cytokines’ concentration in the specific time points. We have added this information in the results section (lines 278-281)

4. 338: Why is it significant that HERV-W is induced, but not HERV-K?

Authors’ Answer: Our findings are interesting since supports that HERV-W play a role in the muscle repair physiology and therefore might be upregulated in this condition, while the newest and most active family- HERV-K (HML-1-10, also analyzed here), is not. This is also critical since the expression of one family but not others is not occurring as a consequence of epigenetic changes after cellular stress, caused by local inflammation, but rather, the HERV regulation may be intricately linked to a very specific role in muscular repair. And this make sense since syncytin-1 is responsible for fusogenic events commonly observed in different cell types formation, such as syncytiotrophoblast[12] and osteoclasts[15,16]. Therefore, since HERV-W is upregulated in a particular exercised individuals, it might be significant to identify this difference. We have highlighted this in the discussion session (lines 354-361)

5. 480-88: The present study presents interesting information, but adding protein analysis would strengthen your results. Also, if possible, it would be even better if you could follow up with the study participants. Hence, I would like to return your paper for major revisions.

Authors’ Answer: We agree with the editor, in fact the protein analysis would strengthen our findings and it was discussed as a limitation of the study (lines 485-486) However, there is a lack of data regarding the modulation of HERVs expression in acute physical exercise. We understand that this modulation might change in the next days or weeks, and different scenario might appear, this should certainly be the focus for next studies, we have also brought this matter in the discussion section (lines 491-497)

# Reviewer 1

We have omitted the reviewer`s response to Plos ONE questions for better reading and we have focused on the major and minor points of the reviewer

1 - I think it would be better to use induction to explain how the expression of HERV-W is regulated by cytokines since you are saying that muscle repair is promoted by changes in the cytokine HERV-W, it would be better to explain what has been revealed about the relationship between cytokines and HERV-W.

Authors’ Answer: We would like to thank you for the comment and, as suggested, we added a new paragraph in the "Discussion" section in order to improve the understanding of the relationship between cytokines and HERV-W, as presented below.

In the literature, it has been observed that the HERV-W family is not the only one capable of triggering the production of certain proinflammatory cytokines, such as IL-1β, IL-6, and TNF-α. This occurs through its interaction with Toll-like receptors (TLRs), particularly TLR4 and CD14. These interactions indicate that HERV transactivation can act as a factor fueling inflammation. Moreover, HERVs can activate the immune response, particularly innate immunity, leading to uncontrolled inflammation that may progress toward chronic inflammation[17], or even some HERV Env proteins, particularly derived by HERV-H, HERV-K, and HERV-FRD (Syncytin-2), could be involved in immunosuppressive aspects [18], by induction of the anti-inflammatory cytokine IL-10 [19,20]. Additionally, it has been demonstrated that TNF-α is capable of increasing the RNA expression of HERV-H, HERV-K, and HERV-W through TNF-α receptor signaling[21]. Together, these findings highlight a close interplay between HERVs and inflammation, with HERVs exhibiting both proinflammatory and immunosuppressive roles. Lines (379-392).

2- It would be helpful to provide the rationale for why the time point was set at 45 minutes. I think signals of muscle repair from exercise should be visible for several days, but it would be helpful to provide evidence that it is best to see them after 45 minutes with your chosen training method.

Authors’ Answer: We would like to thank you for the comment and, first of all, we agree that muscle repair signals from exercise can be assessed for several hours/days. In addition, as suggested, we added a new paragraph in the "Discussion" section in order to clarify the reasons for assessing the study parameters after 45 minutes of the exercise session ends, as presented below. 

It is worth mentioning that, in agreement with the literature, not only an acute exercise session is able to elicit several gene transcriptions but also it was reported that after 45 minutes more than 50% of these genes can be upregulated [22]. Specifically, resistance training can activate important signaling pathways 45 minutes after a bout of this exercise type as compared with rest. Moreover, it was reported that, at this time point, both restoration of tissue homeostasis and integrity, and also protein synthesis and degradation are still being regulated [23] and, interestingly, the human skeletal muscle is more sensitive to acute resistance training in the trained than untrained individuals, which reinforce the adaptive process that occurs in response to repeated stimulation [24]. These pieces of information may help us to understand the great changes that can be observed between 45 to 60 minutes after a bout of exercise in some molecules released in response to muscle contracting, such as IL-6 and IL-10 [25] (Lines 362-373)

3- I think it would be better to clearly state whether the exercises in this method are eccentric or concentric, if possible. This is because I think there is a distinction between eccentric and concentric movements when discussing their effects.

Authors’ Answer: We thank the reviewer for the observation, we have added in the methods the characterization of the performed exercises in the methods section (Lines 162-164). We have also discussed this matter in the discussion session as presented below:

Corroborating this last suggestion, it was suggested that eccentric exercise does not necessarily elicit a larger increment of plasmatic IL-6 levels than concentric “nondamaging” muscle contractions, which not only shows that the elevation of systemic IL-6 is not closely associated with muscle damage but also eccentric exercise can lead to a delayed peak, as well as a slow dropped of this myokine in the recovery period [26] (Lines 418-423).

Specifically in the physical exercise context, it was demonstrated that TNF-α levels (both the mRNA in skeletal muscle and protein in the circulation) are, acutely, most prominently increased after the eccentric exercise than after concentric exercise [27] (Lines 429-432).

4- Syncytin-1 is mentioned in the introduction and discussion, but if it is important for the hypothesis of MREV-W expression and muscle repair in this experiment, it would be better to show data on the expression of syncytin-1 as well. think. That will be more persuasive.

Authors’ Answer: We thank the reviewer for this observation and we understand the reviewer concern. In fact, Syncytin-1 encoded by a particular HERV-W env gene, in the chromosome 7q21.2. It was showed that other loci of HERV-W envelope, homologue to Syncytin-1, are also overexpressed under infl

---

## [Decision Letter · Decision Letter 1]

5 Apr 2024

PONE-D-24-05279R1Acute strength exercise training impacts differently the HERV-W expression and inflammatory biomarkers in resistance exercise training individualsPLOS ONE

Dear Dr. Nali,

Thank you for submitting your manuscript to PLOS ONE. After careful consideration, we feel that it has merit but does not fully meet PLOS ONE’s publication criteria as it currently stands. Therefore, we invite you to submit a revised version of the manuscript that addresses the points raised during the review process.

**Reassessment of PLOS paper PONE-D-24-05279R1**

The quality of the paper has improved, and most of the questions have been answered satisfactorily. There is just one point left that I think is questionable. These are just minor details and can be corrected very easily.

Lines 74-75: HERVs are fossil viruses that have infected our ancestors millions of years ago.

This is questionable. Other researchers think they might come from living cells. I suggest to leave out this sentence.

Lines 79-84:

It is utmost of importance to point out that, during evolution, HERVs were exposed to several mutation events, and as a result, there is few complete genomic sequences of the classical retroviral genome, since most of their genes are isolated and distributed within the genome, or even many of them are silenced stop codons within viral genes, which lead to the active replication of HERVs does not occur [14], even though HERVs’ protein is still expressed and the virion may be formed [15–18].

It would be more concise to say: “HERVs were exposed to several mutation events, and as a result, there is few complete genomic sequences of the classical retroviral genome…”

We look forward to receiving your revised manuscript.

Kind regards,

Matthew Cserhati, Ph.D

Academic Editor

PLOS ONE

Reviewers' comments:

Reviewer's Responses to Questions

**Comments to the Author**

1. If the authors have adequately addressed your comments raised in a previous round of review and you feel that this manuscript is now acceptable for publication, you may indicate that here to bypass the “Comments to the Author” section, enter your conflict of interest statement in the “Confidential to Editor” section, and submit your "Accept" recommendation.

Reviewer #1: (No Response)

2. Is the manuscript technically sound, and do the data support the conclusions?

Reviewer #1: Yes

3. Has the statistical analysis been performed appropriately and rigorously? 

Reviewer #1: Yes

4. Have the authors made all data underlying the findings in their manuscript fully available?

Reviewer #1: Yes

5. Is the manuscript presented in an intelligible fashion and written in standard English?

Reviewer #1: Yes

6. Review Comments to the Author

Reviewer #1: Thank you for answering for my comments in a previous round of review. To make the content more persuasive, when discussing Syncytin-1, the expression level of Syncytin-1 should be shown.

7. PLOS authors have the option to publish the peer review history of their article (what does this mean?). If published, this will include your full peer review and any attached files.

Reviewer #1: No

---

## [Author Response · Author response to Decision Letter 1]

5 Apr 2024

April, 5th 2024 

Professor Mathew Matthew Cserhati, Ph.D

Academic Editor, Plos ONE

Dear Professor,

We are pleased for the evaluation of our paper by the editor in this second round of review. In this letter we bring answers to all the questions raised by them. We are also submitting the reviewed version of the manuscript for your consideration.

Editor comments:

Lines 74-75: HERVs are fossil viruses that have infected our ancestors millions of years ago.

This is questionable. Other researchers think they might come from living cells. I suggest to leave out this sentence.

Authors’ Answer: We understand the editor’s concern, and we have removed this sentence as suggested. 

Lines 79-84:

It is utmost of importance to point out that, during evolution, HERVs were exposed to several mutation events, and as a result, there is few complete genomic sequences of the classical retroviral genome, since most of their genes are isolated and distributed within the genome, or even many of them are silenced stop codons within viral genes, which lead to the active replication of HERVs does not occur [14], even though HERVs’ protein is still expressed and the virion may be formed [15–18].

It would be more concise to say: “HERVs were exposed to several mutation events, and as a result, there is few complete genomic sequences of the classical retroviral genome…”

Authors’ Answer: We agree with the editor, in order to make the sentence more concise we have rewritten the sentence as follows: It is utmost of importance to point out that, during evolution, HERVs were exposed to several mutation events, and as a result, there are few complete genomic sequences of the classical retroviral genome, which lead to the active replication of HERVs not occurring [14], even though HERVs’ protein is still expressed and the virion may be formed [15–18].

Reviewer 1 comments

Thank you for answering for my comments in a previous round of review. To make the content more persuasive, when discussing Syncytin-1, the expression level of Syncytin-1 should be shown.

Authors’ Answer: We are pleased by the review and we are glad that our answers and this new version have reached that reviewer’s expectations.

We hope that with this new version may reach the editor’s expectations. Please, do not hesitate if you have further questions.

Sincerely,

Luiz Henrique da Silva Nali

---

## [Editor Report · Decision Letter 2]

29 Apr 2024

PONE-D-24-05279R2Acute strength exercise training impacts differently the HERV-W expression and inflammatory biomarkers in resistance exercise training individualsPLOS ONE

Dear Dr. Nali,

Thank you for submitting your manuscript to PLOS ONE. After careful consideration, we feel that it has merit but does not fully meet PLOS ONE’s publication criteria as it currently stands. Therefore, we invite you to submit a revised version of the manuscript that addresses the points raised during the review process.

We look forward to receiving your revised manuscript.

Kind regards,

Matthew Cserhati, Ph.D

Academic Editor

PLOS ONE

Journal Requirements:

**Additional Editor Comments:**

Dear Dr. Nali,

I do not know why my colleagues at PLOS are so slow about adding your edited document. Let me just send the paper back to you for minor revisions, and please upload the edited version of the paper.

Sorry for being so slow about this.

Remember, the sentence in question is on lines 79-84:

Instead of:

It is utmost of importance to point out that, during evolution, HERVs were exposed to several mutation events, and as a result, there is few complete genomic sequences of the classical retroviral genome, since most of their genes are isolated and distributed within the genome, or even many of them are silenced stop codons within viral genes, which lead to the active replication of HERVs does not occur [14], even though HERVs’ protein is still expressed and the virion may be formed [15–18].

write this:

HERVs were exposed to several mutation events, and as a result, there is few complete genomic sequences of the classical retroviral genome, since most of their genes are isolated and distributed within the genome, or even many of them are silenced stop codons within viral genes, which lead to the active replication of HERVs does not occur [14], even though HERVs’ protein is still expressed and the virion may be formed [15–18].

Thanks, Matthew Cserhati

---

## [Author Response · Author response to Decision Letter 2]

29 Apr 2024

April, 29th 2024 

Professor Mathew Matthew Cserhati, Ph.D

Academic Editor, Plos ONE

Dear Professor,

We are uploading the new version of the manuscript with the suggested reviewed sentence.

Sincerely,

Luiz Henrique da Silva Nali

---

## [Editor Report · Decision Letter 3]

1 May 2024

Acute strength exercise training impacts differently the HERV-W expression and inflammatory biomarkers in resistance exercise training individuals

PONE-D-24-05279R3

Dear Dr. Nali,

We’re pleased to inform you that your manuscript has been judged scientifically suitable for publication and will be formally accepted for publication once it meets all outstanding technical requirements.

Kind regards,

Matthew Cserhati, Ph.D

Academic Editor

PLOS ONE

Additional Editor Comments (optional):

Congratulations!